# Adversarial Bandits with Corruptions

**Lin Yang and Mohammad H. Hajiesmaili**
University of Massachusetts Amherst
{linyang,hajiesmaili}@cs.umass.edu

**M. Sadegh Talebi**
University of Copenhagen
m.shahi@di.ku.dk

**John C. S. Lui and Wing S. Wong**
Chinese University of Hong Kong
cslui@cse.cuhk.edu.hk,wswong@ie.cuhk.edu.hk

## Abstract

This paper studies adversarial bandits with corruptions. In the basic adversarial bandit setting, the reward of arms is predetermined by an adversary who is oblivious to the learner's policy. In this paper, we consider an extended setting in which an attacker sits in-between the environment and the learner, and is endowed with a limited budget to corrupt the reward of the selected arm. We have two main results. First, we derive a lower bound on the regret of any bandit algorithm that is aware of the budget of the attacker. Also, for budget-agnostic algorithms, we characterize an impossibility result demonstrating that even when the attacker has a sublinear budget, i.e., a budget growing sublinearly with time horizon $T$, they fail to achieve a sublinear regret. Second, we propose `ExpRb`, a bandit algorithm that incorporates a biased estimator and a robustness parameter to deal with corruption. We characterize the regret of `ExpRb` and show that for the case of a known corruption budget, the regret of `ExpRb` is tight.

## 1 Introduction

Multi-armed bandits (MABs) [23] present a powerful online learning framework that is applicable to a broad range of application domains including medical trials, web search advertisement, datacenter design, and recommender systems; see, e.g., [5, 24] and references therein. In the basic MAB problem, in each round a learner pulls an arm (corresponding to selecting an action) from a finite set of arms, and observes the reward associated to the selected arm, but not for the other unselected arms. The goal of the learner is to maximize the rewards accumulated in the course of her interaction. MAB problems are typically categorized into stochastic and non-stochastic (or adversarial) problems depending on how the reward sequences are generated. In stochastic bandits [23, 14], rewards are drawn from fixed but unknown distributions, whereas in non-stochastic bandits [3], no statistical assumption on rewards are made and rewards are arbitrary as if they were generated by an adversary.

Motivated by malicious activities in bandit-related applications such as click fraud via malware [26, 22], fake reviews and ratings in recommender systems [11, 17, 27], and email spam [13, 6], there have been recent effort on studying bandit problems under some notion of *corruption* [12, 28, 16, 18, 9, 15, 8, 19]. In the case of click fraud, for example, botnets maliciously simulate users clicking on an ad to mislead learning algorithms. More specifically, there are some rewards (click rates) associated to each arm (ad), and an attacker (the botnet) corrupts the rewards based on the learner's action. The majority of past efforts, however, are limited to studying stochastic bandits with corruption, either on understanding the vulnerability of existing algorithms and designing attacks [12, 28, 16, 15, 8, 19], or developing algorithms that are robust against corruption [18, 9, 30]. In those works, stochastic patterns are corrupted by an attacker and bandit algorithms strive to be robust against the corruption. A detailed literature review is provided in §A of the supplementary material.

Table 1: Summary of prior literature and this work

| | Stochastic Bandits | | | | Non-stochastic Bandits | | |
| --- | --- | --- | --- | --- | --- | --- | --- |
| Reference | Oblivious | Targeted | Vulnerability | Robustness | Targeted | Vulnerability | Robustness |
| Lykouris *et al.* [18] | ✓ | | ✓ | ✓ | | | |
| Gupta *et al.* [9] | ✓ | | ✓ | ✓ | | | |
| Jun *et al.* [12] | | ✓ | ✓ | | | | |
| Liu *et al.* [16] | ✓ | ✓ | ✓ | | | | |
| **This work** | | ✓ | ✓ | | ✓ | ✓ | ✓ |

In contrast, this paper is the first, to the best of our knowledge, that studies *non-stochastic bandits with corruptions*. In some application domains such as shortest path routing [21] and inventory control problem [7], the reward functions are very complex to model using stochastic bandits, and hence, from a practical perspective, non-stochastic bandits are relevant for such intrinsically involved applications.

A concrete example of non-stochastic bandits with corruptions is the Online Shortest Path Routing (OSPR) problem under the denial of service (DoS) attacks. OSPR is a classic example of MAB problems that has been studied in both stochastic and adversarial settings [4, 21, 25]. And there is also extensive research on routing under DoS attacks, including the recent work [29] focusing on bandit modeling of this scenario. OSPR could be reasonably modeled as non-stochastic bandits when the delays on the links change dynamically in an predictable manner [10], or in situations where the combined distribution of a path including multiple links is difficult to characterize [21]. In this non-stochastic scenario, the DoS attack could be modeled by our bandit with targeted corruptions. Specifically, the DoS attacker can be aware of the selected paths by detecting the transmitted packets over the path and manipulate the latency of the selected path by flooding the path with dummy packets. Also, the budget of the attacker is simply the available resources for the DoS attacker to keep her undetectable. Arguably, none of "non-stochastic bandits" and "stochastic bandits with corruption" models alone would suffice to fully characterize the underlying model here. In addition, this problem is interesting with unique challenges different from stochastic bandits with corruptions and calls for non-trivial algorithm design and regret analysis. Consequently, studying the vulnerability and robustness of non-stochastic bandit algorithms with corruption becomes important as well. We formally define the corruption model in the following.

## 1.1 The Corruption Model

Consider a $K$-armed non-stocahstic bandit, similar to the model in [3], where the rewards are generated by an adversary *obliviously*, namely they are generated before the game starts. At each round $t \in [T]$, the learner selects an arm $I_t \in [K]$ with the *primary reward* $x_{I_t}(t) \in [0, 1]$. In the corruption model, *there is an attacker that sits in-between the environment and the learner, observes the arm chosen by the learner, and corrupts its rewards aiming to mislead the learner to select sub-optimal arms.* More specifically, the attacker manipulates the reward into $\tilde{x}_{I_t}(t) = x_{I_t}(t) - a(t)$, where $a(t) \in [x_{I_t}(t) - 1, x_{I_t}(t)]$ denotes the attack in round $t$. The learner receives $\tilde{x}_{I_t}(t)$ without knowing the original reward $x_{I_t}(t)$. The attacker is aware of the selected arm, and can set the value of $a(t)$ to attack the learner to end up with selecting a sub-optimal arm. Further, similar to existing work on stochastic bandits with corruptions [9, 18], the total budget of the attacker is upper bounded by $\Phi$. The formal statement of the model is given in §2. We emphasize that while considering an *oblivious adversary*, in this model the attacker manipulates the reward adaptively to the learner's chosen arm; hence, the attacker is different from the adaptive adversary in which the rewards is determined right before the learner's action. We refer to this attack as *targeted*. In contrast, the prior literature on stochastic bandits with corruption [18, 9] assume an oblivious attacker who manipulates the rewards before observing the learner's chosen arm. We call these attacks *oblivious*. Last, we refer to the algorithms that are unaware of the existence of the attacker (or its budget) as *attach-agnostic* algorithms, and *attack-aware* algorithms know the attacker and its budget.

## 1.2 Summary of Contributions

In addition to introducing the above non-stochastic bandits with targeted corruptions, this paper investigates the vulnerability of attack-agnostic algorithms and establishes a regret lower bound for

attack-aware algorithms. Then, as the main contribution, this paper presents a robust bandit algorithm in the corrupted setting. Table 1 highlights the high-level contributions of this work as compared to the related literature.

### 1.2.1 Vulnerability and Regret Lower Bound

We first derive an impossibility result for obtaining a sublinear regret for *attack-agnostic* algorithms for non-stochastic bandits with a sublinear attacker. Our results, presented in Theorem 1 in §3, show that even when an attacker has a sublinear budget, any attack-agnostic bandit algorithm fails to achieve a sublinear regret. This impossibility result applies to stochastic bandit algorithms with targeted corruptions as well. Our impossibility result does not contradict the attack-agnostic algorithms in [18, 9] that develop no-regret algorithms for oblivious attackers.

### 1.2.2 Robust Algorithm Design and Regret Analysis

As the main contribution, in §4, we then propose `ExpRb`, that if aware of $\Phi$, achieves a sublinear regret given sublinear $\Phi$, hence robust. The key ideas of `ExpRb` is to first identify the most vulnerable arms against attacker as a function of selection probabilities; a piece of information that is available to the learner. Then, `ExpRb` constructs a robust estimator that biases (possibly) corrupted reward of the vulnerable arms to mitigate the risk of underestimating the actual reward. Our robust estimator is carefully designed to bias the observed rewards just enough to prevent overestimating the actual reward as well. The impossibility result in Theorem 1 shows that a no-regret algorithm should be attack-aware, which may not be possible in practice. Hence, we adapt a middle-ground approach such that the robustness power of `ExpRb` against corruption is controlled by a robustness parameter $\gamma$, which impacts the design of the robust estimator. Last, in §5, we analyze the regret of `ExpRb` and in Theorem 3 and show that if $\gamma = \Phi$, the regret of `ExpRb` is $O(\sqrt{T} + \Phi \log T)$.

## 2 Preliminaries and Problem Statement

### 2.1 The Classical Adversarial MAB Problem

The adversarial (or non-stochastic, used interchangeably) MAB problem, initially introduced in [3], is a game in which a learner repeatedly chooses an arm from a set $[K] := \{1, \ldots, K\}$ of arms in each round. Let $x_i(t) \in [0, 1]$ denote the reward associated to arm $i \in [K]$ in round $t$. For each $i$, the reward sequence $(x_i(t))_{t \in [T]}$ is determined by an adversary before the game starts.[1] At each round $t \in [T]$, the learner chooses an arm $I_t \in [K]$ and receives $x_{I_t}(t)$ as feedback. The objective of the learner is to devise an arm selection algorithm $\mathcal{A}$ maximizing the cumulative rewards over $T$ steps. The performance of the algorithm $\mathcal{A}$ is measured through the notion of pseudo-regret (regret, for short), which is defined as the difference between the cumulative rewards attained by always taking an optimal static decision (in hindsight) and that of $\mathcal{A}$, i.e.,

$$\text{REGRET}(T, \mathcal{A}) = \max_{i \in [K]} \sum_{t=1}^{T} x_i(t) - \mathbb{E}\left[\sum_{t=1}^{T} x_{I_t}(t)\right], \tag{1}$$

where the expectation is taken with respect to possible internal randomizations of $\mathcal{A}$. The `Exp3` algorithm [3] is the first proposed algorithm achieving a regret of $O(\sqrt{KT \log(K)})$ for the classical adversarial bandit problem described above, and whose advent has led to several other learning strategies with improved regret bounds or applicable to more general settings; see, e.g., [1, 2] and references in [24]. In the following, we introduce a new extended model in which an attacker sits in-between the environment and the learner and corrupts the reward of the selected arm.

### 2.2 Adversarial Bandits with Corruptions

Consider an adversarial bandit problem, where an adversary and an attacker with more powerful ability to manipulate the reward coexist. Similarly to the classical adversarial bandit described above,

the adversary determines the reward in an arbitrary way prior to the first round. In runtime, after the learner commits to an arm, the attacker is able to corrupt the reward of the selected arm $I_t$, and the learner receives the corrupted reward. Specifically speaking, the attacker manipulates the reward $x_{I_t}(t)$ of the selected arm $I_t$ into

$$\tilde{x}_{I_t}(t) = x_{I_t}(t) - a(t), \qquad a(t) \in [x_{I_t}(t) - 1, x_{I_t}(t)], \tag{2}$$

where $a(t)$ is the *injected corruption* (or corruption, for short) at round $t$. Note that the feasible range of corruption at round $t$ implies $\tilde{x}_{I_t}(t) \in [0, 1]$. The learner receives $\tilde{x}_{I_t}(t)$ without knowing the original reward $x_{I_t}(t)$ or the corruption $a(t)$.

The value of $a(t)$ in Eq. (2) determines the design space of the attacker in each round to mislead the learner to end up with selecting a suboptimal arm. However, we assume that the attacker is endowed with a predetermined corruption budget. Let $\Phi(T)$ represent the budget of the attacker, so that cumulative exerted corruption (magnitude-wise) over all rounds must satisfy $\sum_{t=1}^{T} |a(t)| \leq \Phi(T)$. We further refer to such an attacker as a $\Phi(T)$-*attacker*. Clearly, the performance of algorithms degrades more for larger values of $\Phi(T)$. Hereafter, we denote $\Phi := \Phi(T)$ for brevity.

In the following definition, we formally characterize the notion of robustness of a bandit algorithm against corruptions.

**Definition 1** *An algorithm $\mathcal{A}$ is said to be $\Phi$-robust if $\text{REGRET}(T, \mathcal{A}) = \widetilde{O}(\sqrt{T} + \Phi)$ against any $\Phi$-attacker, where the $\widetilde{O}(\cdot)$ notation hides multiplicative terms that are poly-logarithmic in $T$.*

We finally turn to introducing the notion of regret for the adversarial bandits with corruptions. The regret of the algorithm $\mathcal{A}$ is defined as

$$\text{REGRET}(T, \mathcal{A}) = \max_{i \in [K]} \sum_{t=1}^{T} x_i(t) - \mathbb{E}\left[\sum_{t=1}^{T} \tilde{x}_{I_t}(t)\right], \tag{3}$$

where the second term in the right-hand side corresponds to the expected return in terms of corrupted values. We remark that it is plausible to consider a slightly different version of the attack model, which only changes the *observation* of the learner without changing the *actual* accrued reward. In this case, the definition of regret coincides with that in Eq. (1). Our regret analysis for the notion of regret in Eq. (3) could be straightforwardly applied to that of Eq. (1). Details in Remark 5.1 in §5. Unless stated otherwise, the term "regret" in this paper refers to the notion formalized in Eq. (3).

**Remark 2.1** *We mention that there is growing literature on oblivious attack models for stochastic bandit problems; see, e.g., [18, 9]. These papers target at a middle ground of a mixed stochastic and adversarial model that aim to achieve the best of both worlds. Different from these works, our work focuses on targeted attack models for adversarial bandits, since an oblivious attacker can be intrinsically captured in the basic setting of adversarial bandits.*

**Remark 2.2** *There is a rich literature on non-stochastic bandits with adaptive adversaries [24], where the adversary is able to see the past actions of the learner and determines the reward right before the current action. The attacker in our model is more powerful than an adaptive adversary, since it observes the action of the learner and perturbs the reward before revealing it to the learner.*

## 3 Vulnerability and Regret Lower Bound

In this section, we present a regret lower bound for *attack-agnostic* algorithms, i.e., algorithms that are unaware of the existence of an attacker.

We begin with the following theorem establishing a linear regret for *attack-agnostic* algorithms against a $\Phi$-attacker with $\Phi = o(T)$:

**Theorem 1** *Consider an attack-agnostic bandit algorithm $\mathcal{A}$ satistfying the following property: For any two-armed problem instance, the expected regret of $\mathcal{A}$ is $O(\sqrt{T})$. Then, for any $\varepsilon \in (0, \frac{1}{8})$, there exists a $\Phi$-attacker with $\Phi = O(T^{1/2+2\epsilon})$ such that the regret of $\mathcal{A}$ (without knowing the attack) is $\Omega(T^{1-\varepsilon})$ with high probability.*

The above theorem demonstrates an impossibility result for attack-agnostic bandit algorithms to achieve a sublinear regret. We stress that this result is applicable to stochastic MABs with targeted corruptions as well. We however stress that Theorem 1 has no conflict with the results in [18, 9] where robust corruption-agnostic algorithms designed for stochastic MABs with *oblivious* corruption. In fact, the proof of this theorem, provided in §B in the supplementary, constructs an instance of a stochastic bandit problem and considers the setting that the reward on each arm is subject to a fixed and unknown distribution. In order to attain a sublinear regret, the learning algorithm can only sample a "sub-optimal" arm for sublinear number of times. Otherwise, the learning algorithm fails to attain a sublinear regret even without attacks. Thus, the attacker can mislead the algorithm by manipulating the reward on the optimal arm for sublinear number of times. Consequently, the optimal arm is sampled for only sublinear number of times, and the regret of any attack-agnostic bandit algorithm can thus be made arbitrarily close to linear (by choosing small enough $\varepsilon$).

As a concrete example, in the following we show that the classic $\texttt{Exp3}$ algorithm[2] cannot achieve a sublinear regret against an $O(\sqrt{T})$-corrupted attacker.

**Corollary 2 (Vulnerability of $\texttt{Exp3}$)** $\varepsilon \in (0, \frac{1}{8})$. *There exists a $\Phi$-attacker with $\Phi = O(T^{1/2+2\epsilon})$ such that the regret under $\texttt{Exp3}$ is $\Omega(T^{1-\varepsilon})$.*

Theorem 1 demonstrates that to develop a robust algorithm for non-stochastic bandits with corruptions, it is inevitable to provide the algorithm with the information of the existence of the attacker. We call these algorithms attack-aware algorithms. However, it remains open whether the knowledge of the attacker's budget is necessary to attain a sublinear regret.

## 4 The $\texttt{ExpRb}$ Algorithm

In this section, we propose $\texttt{ExpRb}$, a bandit algorithm that is robust to corruption from a targeted attacker. The logical flow of $\texttt{ExpRb}$ follows the rationality of the $\texttt{Exp3}$ algorithm with an additional novel biased estimator to make the algorithm robust against corruption. In round $t \in [T]$, $\texttt{ExpRb}$ draws arm $I_t$ according to the following distribution

$$p_i(t) = (1 - \eta) \frac{w_i(t-1)}{\sum_{j=1}^{K} w_j(t-1)} + \frac{\eta}{K}, \quad i \in [K], \tag{4}$$

which is a weighted combination (parameterized by $\eta \in (0, 1]$) of a uniform distribution and a weighted distribution determined by the weights $w_i(t-1)$ maintained for each arm. The weight parameter $w_i(t)$ is defined for each arm with initial values of 1. The intuition behind selecting this mixed distribution is to make sure that all arms are chosen [3].

Once the algorithm selects arm $I_t$ the estimated reward is calculated as follows.

$$\hat{x}_i(t) = \mathbb{1}_{\{I_t = i\}} \frac{\tilde{x}_i(t) + \delta(t)}{p_i(t)}, \quad i \in [K], \tag{5}$$

where $\mathbb{1}_A$ denotes the indicator function of an event $A$, and where $\delta(t)$ is a compensate variable explained in details in §4.1. Finally, the algorithm updates the weight of the various arms as

$$w_i(t) = w_i(t-1) \exp\left(\eta \hat{x}_i(t)/K\right), \quad i \in [K]. \tag{6}$$

In the next section, we explain the details of the robust estimator as the key novelty of the $\texttt{ExpRb}$ algorithm.

### 4.1 Robust Estimator and Intuitions

Once the arm $I_t$ is selected the main step of $\texttt{ExpRb}$ toward robustification of the observed reward $\tilde{x}_{I_t}(t)$ begins. The high-level idea of robustification is two-fold: (i) we introduce a compensate variable $\delta(t)$ to augment the estimated reward of the selected arm and mitigate the risks of underestimation and overestimation of the actual reward; and (ii) we introduce a robustness parameter $\gamma$ that could be tuned based on the budget of the attacker, to determine the design space of learner in biasing the estimated reward.

**Algorithm 1** The ExpRb Algorithm
---
1: **Initialization:** $\eta \in (0, 1]$, robustness parameter $\gamma$, $w_i(0) = 1$ and $q_i(0) = 1$ for all $i \in [K]$
2: **for** $t = 1$ to $T$ **do**
3:  Set
$$p_i(t) = (1 - \eta)\frac{w_i(t-1)}{\sum_{j=1}^{K} w_j(t-1)} + \frac{\eta}{K}, \quad i \in [K]$$
4:  Draw arm $I_t$ randomly according to the probabilities $p_1(t), \ldots, p_K(t)$
5:  Observe reward $\tilde{x}_{I_t}(t)$
6:  Set $\delta(t) = 0$
7:  **if** $p_{I_t}(t) < q_{I_t}(t-1)$ **then**
8:   Set $\delta(t) = \min\{\gamma(1 - p_{I_t}(t)/q_{I_t}(t-1)), 1\}$
9:   Update $q_{I_t}(t) = \begin{cases} \max\{p_i(t), (1 - 1/\gamma)q_i(t-1)\} & i = I_t \\ q_i(t-1) & i \neq I_t \end{cases}$
10: **end if**
11: Set the reward estimates
$$\hat{x}_i(t) = \mathbb{1}_{\{I_t = i\}}\frac{\tilde{x}_i(t) + \delta(t)}{p_i(t)}, \quad i \in [K]$$
12: Update the weights
$$w_i(t) = w_i(t-1)\exp\left(\eta\hat{x}_i(t)/K\right), \quad i \in [K]$$
13: **end for**
---

Now, we proceed to explain the details of the robust estimator. As Eq. (5) indicates, if $p_{I_t}(t)$, the selection probability of the selected arm $I_t$, is small, the attacker is able to greatly impact the estimated reward of $I_t$ with small corruption. In other words, when the selection probability for the selected arm is small, the required budget for the attacker to trick the learning algorithm to "underestimate" the arm is also small. This leads us to set the value of compensate variable as a function of selection probability. However, the learner should be able to track the historical evolution of compensate variable for each arm to prevent "overestimation" of the corruption. Hence, we initiate an auxiliary variable $q_i(0) = 1, i \in [K]$, to record the smallest selection probability of each arm (if chosen) so far. The value of compensate variable is set as follows.

$$\delta(t) = \min\left\{\gamma\left(1 - \frac{p_{I_t}(t)}{q_{I_t}(t-1)}\right), 1\right\}. \tag{7}$$

The algorithmic nuggets of setting the compensate variable are as follows: (i) as in Line 7 of ExpRb, $\delta(t)$ is set only when $p_{I_t}(t) < q_{I_t}(t-1)$, since otherwise, the algorithm has already biased the estimated reward of $I_t$ in previous rounds; (ii) $\delta(t)$ is capped to at most 1, since the value of $a(t)$, i.e., the attacker's corruption, is at most 1; (iii) $\delta(t)$ is a function of $\gamma$ that determines how much bias is required; $\gamma$ has a direct relationship to the budget of attacker, i.e., the greater the budget of the attacker, the greater the robustness parameter $\gamma$; and last (iv) the larger the difference between $p_{I_t}(t)$ and $q_{I_t}(t-1)$, the greater the $\delta(t)$. And finally, we update $q_i, i \in [K]$ to either $p_{I_t}(t)$ (once the first term in Eq. (7) is active) or the value of $p_{I_t}(t)$ at which $\gamma(1 - p_{I_t}(t)/q_{I_t}(t-1)) = 1$, representing the second term in Eq. (7) in which $\delta(t) = 1$. More compactly, we have

$$q_{I_t}(t) = \max\left\{p_{I_t}(t), \left(1 - 1/\gamma\right)q_{I_t}(t-1)\right\}. \tag{8}$$

The running time of ExpRb is similar to Exp3 which is $O(K)$. The pseudocode of ExpRb is summarized as Algorithm 1.[3] Last, it is worth noting that the idea of compensate variable (a.k.a. biased estimator) has been used for a variety of reasons in the non-stochastic bandits, e.g., in Exp3.P [3] and Exp3.IX [20] the idea of *biased reward-estimates* is leveraged to achieve improved high-probability regret bounds for non-stochastic bandits. Although the high-level idea of "robust estimator" is the same, our design in this work is to make the algorithm robust against corruption.

**Remark 4.1** *We remark that [31] presents the* `Tsallis-INF` *algorithm for the so-called 'best of both worlds' setting.* `Tsallis-INF` *is shown to be robust to adversarial corruptions not only in stochastic bandits but also in a class of adversarial bandits with* stochastically-constrained adversaries*; we refer to Corollary 8 in [31] for the corresponding regret bound of* `Tsallis-INF` *for such adversarial bandits with corruptions. As such,* `Tsallis-INF` *is guaranteed to achieve a sublinear regret in a restricted class of adversarial problems with corruptions. In contrast to [31], in this paper we consider adversarial bandits with corruptions with no such restrictions. However, we would like to note that when applied to the adversarial bandits with corruptions with stochastically-constrained adversaries,* `Tsallis-INF` *is expected to attain a tighter regret bound and without requring the knowledge of* $\Phi$*.*

## 5 Regret Analysis

Finally, we analyze the regret of `ExpRb`, and specifically demonstrate it matches the lower bound (up to a logarithmic factor) for the case where the corruption budget is upper bounded.

### 5.1 Summary and Highlights of the Results

The main result is summarized in the following theorem.

**Theorem 3** *The regret under* `ExpRb`*, when it is run with parameters* $\gamma = \Phi$ *and* $\eta = O(\sqrt{(K \log K)/T})$*, satisfies*

$$\text{REGRET}(T, \texttt{ExpRb}) \leq O\left(\sqrt{K \log KT} + K\Phi \log T\right). \tag{9}$$

The above theorem asserts that the regret upper bound of `ExpRb` scales as $\widetilde{O}(\sqrt{T} + \Phi)$. In view of Definition 1, this implies that `ExpRb` is $\Phi$-robust.

**Remark 5.1** *The result in Theorem 3 uses the modified definition of regret in Eq. (3), where the attacker corrupts the actual reward observed by the learner. However, this result can be straightforwardly translated to the original definition of regret in Eq. (1), where the attacker only manipulates the observations of the learner (i.e., feedback), not her actually accrued rewards. A closer look reveals that the difference between the notions of regret in Eq. (1) and (3) is always upper bounded by* $\Phi$*, which does not dominate the regret upper bound of Theorem 3.*

In the following, we proceed to highlight the key steps to prove the regret result in Theorem 3.

### 5.2 Regret Analysis of `ExpRb`

The full proof of the theorem appears in §C of the supplementary material. We split the regret analysis of `ExpRb` into two parts. First, we analyze the properties of the robust estimator of `ExpRb` as a function of the robustness parameter $\gamma$. These properties then is further applied to analyze the regret of `ExpRb` with respect to $\gamma$ and $\Phi$.

Recall that the robustness parameter $\gamma$ impacts the amount of compensate variable $\delta(t)$ in `ExpRb`. We first characterize an upper bound on the cumulative amount of compensate variable with respect to $\gamma$ in Lemma 4. This result could be interpreted as an upper bound on "overestimation" of rewards. Then, in Lemma 5, we derive a lower bound on the difference between the expected value of the cumulative estimated rewards of arms in `ExpRb` and the actual rewards of the arms. This result could be represented as a lower bound on the "underestimation" of rewards.

The following lemma provides an upper bound on the cumulative compensate variable $\delta(t)$:

**Lemma 4** *Under* `ExpRb`*, we have:* $\sum_{t=1}^{T} \delta(t) \leq \gamma K \log(K/\eta)$*.*

This result provides an upper bound for the cumulative value of compensate variable, which is $O(\gamma \log(1/\eta))$. The proof of this result follows by re-expressing the value of $\delta(t)$ as a function of

$\gamma$ and the auxiliary parameter $q_i$, and then applying straightforward calculus to derive the bound. Details in §C in the supplementary.

The following result characterizes the performance of the robust estimator as a function of $\gamma$ and $\phi$.

**Lemma 5** *When* `ExpRb` *is run with* $\gamma \geq \Phi$ *against a* $\Phi$-*attacker, we have:*

$$\sum_{t=1}^{T} \hat{x}_i(t) \geq \sum_{t=1}^{T} x_i/p_i(t), \quad \forall i \in [K].$$

This implies that when $\gamma \geq \Phi$, i.e., the robustness parameter is large enough to be able to compensate the corruption, the estimator can effectively avoid underestimation, thus guaranteeing that $\sum_{t=1}^{T} \mathbb{E}[\hat{x}_i(t)] \geq \sum_{t=1}^{T} x_i(t)$.

We are ready to sketch the proof of Theorem 3. The proof of Theorem 3 follows similar steps as in the proof of `Exp3` in [3]. We stress, however, that the proof here relies on more involved steps as one has to take into account the impact of compensate variable $\delta(t)$ on the final regret. By applying similar analysis for the proof of `Exp3`, we have

$$\mathbb{E}\left[\sum_{t=1}^{T} \hat{x}_i(t)\right] - \mathbb{E}\left[\sum_{t=1}^{T} \tilde{x}_{I_t}(t)\right] \leq (e^2 - 1)\eta T + \frac{K \log K}{\eta} + \mathbb{E}\left[\sum_{t=1}^{T} \delta(t)\right], \ i \in [K].$$

Compared to the basic setting, our algorithm introduces an additional term $\sum_{t=1}^{T} \mathbb{E}[\delta(t)]$, which corresponds to the long-term sum of the compensate variable. Lemma 4 implies that the sum of the compensate variable is upper bounded by $O(\gamma K \log(K/\eta))$. In addition, in Lemma 5, we have characterized an upper bound on the difference between the cumulative reward $\sum_{t=1}^{T} x_i(t)$ and the estimation $\sum_{t=1}^{T} \mathbb{E}[\hat{x}_i(t)]$. Finally, applying the upper bounds in Lemma 5 concludes the proof of Theorem 3. A detailed proof is given in §C in the supplementary.

## 6   Concluding Remarks

Motivated by the recent interests in making the online learning algorithms robust against manipulation attacks, this paper studied non-stochastic multi-armed bandit problems with targeted corruptions. It first showed that under targeted corruptions, existing attack-agnostic algorithms for non-stochastic bandits, e.g., `Exp3`, are vulnerable against targeted corruptions with limited budget, and fail to achieve a sublinear regret. Second, it proposed `ExpRb`, as a robust algorithm against targeted corruptions and characterized its regret as a function of a parameter that determines the robustness budget of the algorithm against targeted corruptions. The regret analysis shows that if the corruption budget is sublinear and `ExpRb` is aware of this budget, it achieves a sublinear regret. While there are several recent studies that focus on stochastic MAB problems with corruptions, to the best of our knowledge, this paper is the first that tackles non-stochastic MABs with targeted corruptions.

## 7   Broader Impacts

Our work fits within the broad direction of research concerning safety issues in AI/ML at large. With the recent radical advances in machine learning, ML-assisted decision making is fast becoming an intrinsic part of the design of systems and services that billions of people around the world use every day. And not surprisingly, investigating the vulnerability of existing learning models and robustness against manipulation attacks are becoming critically important in the light of *trustworthy learning paradigm*. Hence, there has been a surge of interest in making learning models that are robust against adversarial attacks for both applied ML such as supervised learning and deep learning, and theoretical ML such as reinforcement learning and multi-armed bandits. This is critically important for society, since the ML algorithms are being adopted more and more in safety-critical domains across sciences, businesses, and governments that impact people's daily lives. Last, we see no ethical concerns related to this paper.

## Acknowledgments and Disclosure of Funding

Lin Yang and Wing Shing Wong acknowledge the support from Schneider Electric, Lenovo Group (China) Limited and the Hong Kong Innovation and Technology Fund (ITS/066/17FP) under the HKUST-MIT Research Alliance Consortium. Mohammad Hajiesmaili's research is supported by NSF CNS-1908298. The work of John C.S. Lui is supported in part by the GRF 14201819. Sadegh Talebi's research is supported by Department of Computer Science, University of Copenhagen.

## Footnotes

[1]Some literature consider *loss formulation* of adversarial bandits, where the learner receives a loss $\ell_i(t) \in [0, 1]$ upon choosing arm $i$ in round $t$. Here we consider the reward formulation. We note however that most results for reward formulation can be translated to the corresponding loss formulation via the relation $\ell_i(t) = 1 - x_i(t)$; see [5].

[2]We refer the reader to [3] for the detailed explanation of the $\texttt{Exp3}$ algorithm. The $\texttt{Exp3}$ algorithm, however, could be recovered from Algorithm 1 in this paper by simply setting $\tilde{x}_i(t) = x_i(t)$ and $\delta(t) = 0$ for all $i, t$.

[3]In the paper, the algorithm is presented with fixed parameters with respect to the length of time horizon. One can extend the proposed algorithm to the anytime version by using the doubling trick policy [3].

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
