[Supplementary Material]

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

# A   Related Work

The basic MAB problems have been extensively extended to several other settings. Our literature review, however, is centered on MABs with corruptions. The existing literature on MAB with adversarial corruptions could be categorized based on the corruption model into two categories of *oblivious* and *targeted* corruption models. Further the existing literature could be categorized into those work that study the *vulnerability* of existing algorithms versus those that develop *robust* algorithms against corruptions. Based on these four criteria, Table 1 summarizes the settings of the existing work and this work. In short, the majority of the existing works focus on either oblivious or targeted corruptions for stochastic MAB problems, and this work, to the best of our knowledge, is the first that studies corruption models for non-stochastic bandits.

## A.1   MAB Problems with Oblivious Corruptions

In the oblivious corruption model, an attacker, *oblivious* to the behavior of the bandit algorithm, corrupts the stochastic patterns of some arms in each round. Specifically, this corruption model targets stochastic bandit problems in which the reward of each arm follows a stochastic distribution. The goal of the attacker is to adversarially manipulate the rewards of some arms to trick the algorithm to choose sub-optimal arms. This model targets a middle ground of a mixed stochastic and adversarial model that aims to achieve the best of both worlds. The oblivious corruption model is intrinsically captured in the basic setting of non-stochastic MAB, since the adversary determines the reward in adversarial manner, however, oblivious to the learner's algorithm [3, 5]. In the following, then, we focus on reviewing the related works on stochastic MABs with oblivious corruptions.

Ma *et al.* [19] introduced an attack framework based on a convex optimization formulation that shows by slightly manipulation of the rewards, existing MAB algorithms are highly vulnerable against oblivious corruption models. In [16], the framework has been extended to develop attack strategies to a broad range of stochastic bandit algorithms. Both works, however, focus on designing attack strategies to show the vulnerability of existing algorithms.

In another category [9, 18], the goal is to develop robust algorithms against oblivious corruptions. The high-level idea is to expand the confidence bounds of the existing algorithms to be robust against manipulation attacks on rewards. This setting was first proposed by Lykouris et al. [18] and a sublinear regret algorithm with respect to the corruption budget was proposed. Specifically, the proposed algorithm in [18] achieves the regret of $\tilde{O}(KG \sum_{i \neq i^*} 1/\Delta_i)$, where $K$ is the number of arms, $G$ is the corruption budget, $i^*$ is the optimal arm, and $\Delta_i$ is the gap between $\mu^*$, the expected reward of the optimal arm and $\mu_i$, the expected reward of arm $i$, i.e., $\Delta_i = \mu^* - \mu_i$, and notation $\tilde{O}$ suppresses all dependence on logarithmic terms. This bound is $O(KG)$ times worse than the standard bound achievable by existing algorithms like UCB in uncorrupted setting. This result has been improved to an algorithm with the regret of $O(KG) + \tilde{O}(\sum_{i \neq i^*} 1/\Delta_i)$ in [9]. That is, the new algorithm in [9] attains a regret bound which removes the multiplicative dependence on $G$ in [18] and replace it with an additive term. When the corruption is more powerful, i.e., larger $G$, the reward pattern is more like that of the adversarial model, thereby the performance of the online algorithm is expected to be degraded to fully non-stochastic setting. Last, Zimmert and Seldin [30] study the problem of optimal algorithms for stochastic and adversarial bandits that includes [18, 9] as special case.

## A.2   MAB Problems with Targeted Corruptions

In the targeted corruption model, which is mainly the focus of this paper, *the adversary sits in-between the environment and the learner, observes the selected arm by the learner, corrupts its reward, and the learner just observes the corrupted reward.* That means the corruption policy targets the action of the player, and hence, the corruption is more powerful than the oblivious corruption model. Different from the previous setting, this corruption model could be considered in both stochastic and non-stochastic models.

The prior work in this direction [12, 16] studied the vulnerability of existing stochastic MAB algorithms against targeted corruptions. The authors in [12] design specific targeted attacks with logarithmic budget that hijack two popular stochastic bandit algorithms, i.e., $\epsilon$-greedy and UCB algorithms, by failing to achieve sublinear regret. A more comprehensive vulnerability study is

conducted in [16] where a targeted corruption strategy is proposed that can hijack any stochastic bandit algorithm without knowing the bandit algorithm.

Our work, to the best of our knowledge, is the first that focuses on non-stochastic bandits with targeted corruptions. Similar to [12, 16], it investigates the vulnerability of existing bandit algorithms, however, different from [12, 16] for non-stochastic setting, e.g., `Exp3`. Similar to [9, 18], it develops a robust algorithm, called `ExpRb` for corrupted bandits, however, different form [9, 18] for non-stochastic setting. Last, our analysis on vulnerability is applicable to both stochastic and non-stochastic bandit algorithms.

## B    Proof of Theorem 1

We consider a two-armed bandit problem with Bernoulli arms with means $(\mu_1, \mu_2) = (\frac{1}{2}, \frac{1}{2} + \Delta)$, for some $\Delta$ that we specify later. The rewards of each arm are i.i.d. and the rewards are independent across arms.

Consider a learning algorithm $\mathcal{A}$. Further, consider an attacker, which adds a randomly generated noise with mean $-2\Delta$ to the rewards of the second arm (i.e., the optimal arm) whenever it is selected by the algorithm. Thus, from the learner's view, arm 2 has mean $1/2 - \Delta$. We assume that the adversary has enough budget to do so over $T$ rounds. The algorithm is unaware of the attacker's existence.

Let $N_1$ and $N_2$ denote the number of pulls of arm 1 and arm 2 after $T$ rounds, respectively. Thus, $N_1 + N_2 = T$. The regret on the corrupted problem is $R(T) = \Delta \mathbb{E}[N_2]$. Since we assume that $\mathcal{A}$ attains a regret of at most $O(\sqrt{T})$ for any $T$, we have $\Delta \mathbb{E}[N_2] \leq O(\sqrt{T})$, or $\mathbb{E}[N_2] \leq O(\sqrt{T})/\Delta$. Hence, using $N_1 + N_2 = T$, we get

$$\mathbb{E}[N_1] \geq T - \frac{O(\sqrt{T})}{\Delta},$$

and thus, the regret of $\mathcal{A}$ on the corrupted problem is at least

$$R(T) = \Delta \mathbb{E}[N_1] \geq \Delta T - \frac{O(\sqrt{T})}{\Delta}.$$

Next we find a high probability upper bound on the budget $\Phi$ of the attacker. Observe that $\Phi < N_2$, so we need to find a high probability upper bound on $N_2$. For $X > 0$, we have:

$$X \mathbb{P}(N_2 > X) \leq \mathbb{E}[N_2] \leq \frac{O(\sqrt{T})}{\Delta}$$

Hence, $\mathbb{P}(N_2 < X) \geq 1 - \frac{O(\sqrt{T})}{X\Delta}$ and thus, $\mathbb{P}(\Phi < X) \geq 1 - O(\sqrt{T})/(X\Delta)$. Now choosing $\Delta = T^{-\varepsilon}, \varepsilon \in (0, \frac{1}{8})$ yields

$$R(T) = \Delta \mathbb{E}[N_1] \geq \Delta T - \frac{O(\sqrt{T})}{\Delta} = T^{1-\varepsilon} - O(T^{1/2+\varepsilon}).$$

Hence, $R(T) = \Omega(T^{1-\varepsilon})$. Furthermore, choose $X = T^{1/2+2\varepsilon}$. Then, with high probability, $\Phi \leq O(T^{1/2+2\varepsilon})$. $\square$

## C    Regret Analysis of `ExpRb`: Proof of Theorem 3

Let $T > 1$. For any arm $i$, we let $\mathcal{T}_i \subseteq [T]$ denote the set of time slots where arm $i$ is selected and the selection probability for arm $i$ is lower than all previous ones:

$$\mathcal{T}_i = \left\{ t \in [T] : I_t = i \text{ and } p_i(t) \leq \min_{t' < t : I_{t'} = i} p_i(t') \right\}.$$

We denote the size of $\mathcal{T}_i$ by $N_i$. Alternatively, we may write $\mathcal{T}_i = \{t_i(n), n \in [N_i]\}$. Note that $t_i(n)$, $n = 1, 2, \ldots, N_i$ correspond to the time slots that arm $i$ is selected by `ExpRb` and the maintained probability is updated.

We first provide the following lemmas, which we prove later:

**Lemma 6** *For all $j \in [K]$,*

$$(1 - \eta) \sum_{t=1}^{T} \hat{x}_j(t) \leq \sum_{t=1}^{T} \tilde{x}_{I_t}(t) + \sum_{t=1}^{T} \delta(t) + \frac{\eta}{K} \sum_{t=1}^{T} \sum_{i=1}^{K} \hat{x}_i(t) + \frac{K \log K}{\eta}$$

**Lemma 4 (restated)** *We have:* $\sum_{t=1}^{T} \delta(t) \leq \gamma K \log(K/\eta)$.

Using the above lemmas and taking expectations, we get

$$(1 - \eta) \sum_{t=1}^{T} \mathbb{E}[\hat{x}_j(t)] \leq \sum_{t=1}^{T} \mathbb{E}[\tilde{x}_{I_t}(t)] + \gamma K \log \frac{K}{\eta} + \frac{\eta}{K} \sum_{t=1}^{T} \sum_{i=1}^{K} \mathbb{E}[\hat{x}_i(t)] + \frac{K \log K}{\eta}$$

$$= \sum_{t=1}^{T} \mathbb{E}[\tilde{x}_{I_t}(t)] + \gamma K \log \frac{K}{\eta} + 2\eta T + \frac{K \log K}{\eta},$$

where we used the fact that

$$\mathbb{E}[\hat{x}_j(t)] = \mathbb{E}[\mathbb{E}[p_{I_t}(t)(\tilde{x}_{I_t}(t) + \delta(t))/p_{I_t}(t)|\mathcal{F}_{t-1}]] \leq 2\,,$$

where $\mathcal{F}_{t-1}$ denotes the history of the game up to time slot $t$. We will be using the following lemma to simplify the left-hand side of the inequality:

**Lemma 5 (restated)** *For all $i \in [K]$, and $\gamma \geq \Phi$, we have:* $\sum_{t=1}^{T} \hat{x}_i(t) \geq \sum_{t=1}^{T} x_i(t)/p_i(t)$.

We therefore obtain:

$$\max_j \sum_{t=1}^{T} x_j(t) - \sum_{t=1}^{T} \mathbb{E}[\tilde{x}_{I_t}(t)] \leq \gamma K \log \frac{K}{\eta} + 3\eta T + \frac{K \log K}{\eta}\,.$$

Finally, the proof is completed by setting $\gamma = \Phi$ and $\eta = O(\sqrt{K \log K / T})$. $\qquad\square$

## C.1 Proof of Lemma 6

For $t \geq 1$, denote $W_t := \sum_{i=1}^{K} w_i(t)$. We derive upper and lower bounds on $\log \frac{W_{T+1}}{W_1}$.

**Lower Bound.** Note that $W_1 = K$. Then,

$$\log \frac{W_{T+1}}{W_1} \geq \log \frac{w_j(T+1)}{K} = \frac{\eta}{K} \sum_{t=1}^{T} \hat{x}_j(t) - \log K\,, \tag{10}$$

where $j \in [K]$ is arbitrary.

**Upper Bound.** First observe that $\log \frac{W_{T+1}}{W_1} = \sum_{t=1}^{T} \log \frac{W_{t+1}}{W_t}$. Next we derive an upper bound on $W_{t+1}/W_t$. We have:

$$\frac{W_{t+1}}{W_t} = \sum_{i=1}^{K} \frac{w_i(t)}{W_t} e^{\eta \hat{x}_i(t)/K}$$

$$\leq \sum_{i=1}^{K} \left( \frac{p_i(t) - \eta/K}{1 - \eta} \right) \left( 1 + \frac{\eta}{K} \hat{x}_i(t) + \frac{\eta^2}{2K^2} \hat{x}_i(t)^2 \right)$$

$$= \underbrace{\sum_{i=1}^{K} \frac{p_i(t) - \eta/K}{1 - \eta}}_{=1} + \frac{\eta}{K(1 - \eta)} \sum_{i=1}^{K} p_i(t) \hat{x}_i(t) + \frac{\eta^2}{2K^2(1 - \eta)} \sum_{i=1}^{K} p_i(t) \hat{x}_i(t)^2$$

where in the second line, we have used the inequality $e^z \leq 1 + z + \frac{1}{2} z^2$ valid for all $z > 0$.

Note that $\sum_{i=1}^{K} p_i(t) \hat{x}_i(t) = \tilde{x}_{I_t}(t) + \delta(t)$ and

$$\sum_{i=1}^{K} p_i(t) \hat{x}_i(t)^2 = p_{I_t}(t) \hat{x}_{I_t}(t)^2 = (\tilde{x}_{I_t}(t) + \delta(t)) \hat{x}_{I_t}(t) \leq 2\hat{x}_{I_t}(t) = 2 \sum_{i=1}^{K} \hat{x}_i(t)$$

Hence,

$$\frac{W_{t+1}}{W_t} \le 1 + \frac{\eta}{K(1-\eta)}(\tilde{x}_{I_t}(t) + \delta(t)) + \frac{\eta^2}{K^2(1-\eta)}\sum_{i=1}^{K}\hat{x}_i(t)$$

Taking logarthim from both sides and using the inequality $\log(1+z) \le z$ valid for all $z > -1$, we obtain:

$$\log \frac{W_{t+1}}{W_t} \le \frac{\eta}{K(1-\eta)}(\tilde{x}_{I_t}(t) + \delta(t)) + \frac{\eta^2}{K^2(1-\eta)}\sum_{i=1}^{K}\hat{x}_i(t),$$

which further gives:

$$\log \frac{W_{T+1}}{W_1} = \sum_{t=1}^{T}\log \frac{W_{t+1}}{W_t} \le \frac{\eta}{K(1-\eta)}\sum_{t=1}^{T}\tilde{x}_{I_t}(t) + \frac{\eta}{K(1-\eta)}\sum_{t=1}^{T}\delta(t) + \frac{\eta^2}{K^2(1-\eta)}\sum_{t=1}^{T}\sum_{i=1}^{K}\hat{x}_i(t).$$

Putting the upper and lower bounds together, we obtain: For all $j \in [K]$,

$$\frac{\eta}{K}\sum_{t=1}^{T}\hat{x}_j(t) - \log K \le \frac{\eta}{K(1-\eta)}\sum_{t=1}^{T}\tilde{x}_{I_t}(t) + \frac{\eta}{K(1-\eta)}\sum_{t=1}^{T}\delta(t) + \frac{\eta^2}{K^2(1-\eta)}\sum_{t=1}^{T}\sum_{i=1}^{K}\hat{x}_i(t)$$

which concludes the proof. □

## C.2 Proof of Lemma 4

By the design of `ExpRb`, the value of $\delta(t)$ is set to a non-zero value only when the current selection probability of the selected arm, i.e., $p_{I_t}(t)$ is smaller than $q_{I_t}(t-1)$. Fix an arm $i \in [K]$, and consider time slots $t_i(n)$, $n = 1, 2, \ldots, N_i$, where $i$ is selected. We can show that

$$\delta(t_i(n)) = \gamma \left(1 - \frac{q_i(t_i(n))}{q_i(t_i(n-1))}\right), \quad n = 1, 2, \ldots, N_i. \tag{11}$$

To prove this claim, we consider all possible cases a time slot $t_i(n)$ as follows:

**Case (i):** $p_i(t) \ge q_i(t_i(n-1))$. In this case, $q_i(t_i(n))$ will be set to $q_i(t_i(n-1))$. Then, the value of $\delta(t_i(n))$ from Eq. (11) will be 0, complying with Line 6 of `ExpRb`.

**Case (ii):** $q_i(t_i(n-1)) \ge p_i(t) \ge (1-1/\gamma)q_{I_t}(t_i(n-1))$. Here, $q_i(t_i(n))$ will be set to $p_i(t_i(n))$. Based on Eq. (11), the value of $\delta(t_i(n))$ will be set to $\gamma\left(1 - p_i(t_i(n))/q_i(t_i(n-1))\right)$. This case complies with Eq. (7), since $p_i(t)$ satisfies $\gamma\left(1 - p_{I_t}(t)/q_{I_t}(t_i(n-1))\right) \le 1$.

**Case (iii):** $p_i(t) < (1-1/\gamma)q_{I_t}(t_i(n-1))$. In this case, $q_i(t_i(n))$ will be set to $(1-1/\gamma)q_{I_t}(t_i(n-1))$. In this case, $\delta(t_i(n))$ based on Eq. (11) will be equal to 1. Moreover, when $p_i(t) < (1-1/\gamma)q_{I_t}(t_i(n-1))$, we have $\gamma\left(1 - p_{I_t}(t)/q_{I_t}\right) > 1$, which implies that $\delta(t_i(n))$ complies with Eq. (7).

Putting these together proves the claim in Eq. (11).

$$\begin{aligned}
\sum_{n\in[N_i]}\delta(t_i(n)) &= \sum_{n\in[N_i]}\gamma\left(1 - \frac{q_i(t_i(n))}{q_i(t_i(n-1))}\right) \\
&= \sum_{n\in[N_i]}\gamma\frac{1}{q_i(t_i(n-1))}\left(q_i(t_i(n-1)) - q_i(t_i(n))\right) \\
&\le -\gamma\int_{q_i(t_i(1))}^{q_i(t_i(N_i))}\frac{1}{z}dz = -\gamma\log z\Big|_{1}^{q_i(t_i(N_i))} = -\gamma\log q_i(t_i(N_i)).
\end{aligned}$$

Moreover, by the design of `ExpRb`, $p_i(t) \ge \eta/K$ for all $i$ and $t$, which further implies $q_i(t_i(N_i)) \ge \eta/K$. Hence,

$$\sum_{t\in[T]}\delta(t) = \sum_{i=1}^{K}\sum_{n\in[N_i]}\delta(t_i(n)) \le \gamma K\log(K/\eta),$$

thus completing the proof. □

### C.3 Proof of Lemma 5

Let $i \in [K]$. Due to using compensate variables, the estimation on arm $i$ at time slot $t$ will be increased by $\delta(t)/p_i(t)$. Specifically, by the design of the algorithm, we have

$$\sum_{t \in \mathcal{T}_i} \frac{\delta(t)}{p_i(t)} = \sum_{n \in [N_i]} \frac{1}{p_i(t_i(n))} \left[ \min \left\{ 1, \gamma \left( 1 - \frac{p_i(t_i(n))}{q_i(t_i(n-1))} \right) \right\} \right].$$

To further simplify the above equation, we consider the following two possibilities for a time slot $t_i(n), i \in [N_i]$:

(i) If $\gamma \left( 1 - \frac{p_i(t_i(n))}{q_i(t_i(n-1))} \right) \leq 1$, then $q_i(t_i(n)) = p_i(t_i(n))$ (see Eq. ((8))) and

$$\frac{1}{p_i(t_i(n))} \left[ \min \left\{ 1, \gamma \left( 1 - \frac{p_i(t_i(n))}{q_i(t_i(n-1))} \right) \right\} \right] = \frac{\gamma}{q_i(t_i(n))} \left( 1 - \frac{q_i(t_i(n))}{q_i(t_i(n-1))} \right)$$

$$= \frac{1}{q_i(t_i(n))} \gamma \left( 1 - \frac{q_i(t_i(n))}{q_i(t_i(n-1))} \right) + \left( \frac{1}{p_i(t_i(n))} - \frac{1}{q_i(t_i(n))} \right).$$

(ii) If $\gamma \left( 1 - \frac{p_i(t_i(n))}{q_i(t_i(n-1))} \right) > 1$, according to Eq. ((8)), we have $q_i(t_i(n)) = (\gamma - 1)/\gamma q_i(t_i(n-1))$, so that $\gamma \left( 1 - \frac{q_i(t_i(n))}{q_i(t_i(n-1))} \right) = 1$. Hence,

$$\frac{1}{p_i(t_i(n))} \left[ \min \left\{ 1, \gamma \left( 1 - \frac{p_i(t_i(n))}{q_i(t_i(n-1))} \right) \right\} \right] = \frac{1}{p_i(t_i(n))}$$

$$= \frac{1}{q_i(t_i(n))} + \frac{1}{p_i(t_i(n))} - \frac{1}{q_i(t_i(n))}$$

$$= \frac{\gamma}{q_i(t_i(n))} \left( 1 - \frac{q_i(t_i(n))}{q_i(t_i(n-1))} \right) + \left( \frac{1}{p_i(t_i(n))} - \frac{1}{q_i(t_i(n))} \right).$$

Putting together both cases yields

$$\sum_{t \in \mathcal{T}_i} \frac{\delta(t)}{p_i(t)} = \sum_{n \in [N_i]} \frac{\gamma}{q_i(t_i(n))} \left( 1 - \frac{q_i(t_i(n))}{q_i(t_i(n-1))} \right) + \sum_{n \in [N_i]} \left( \frac{1}{p_i(t_i(n))} - \frac{1}{q_i(t_i(n))} \right) \quad (12)$$

Then, we have

$$\sum_{t \in [T]} \hat{x}_i(t) = \sum_{t \in \mathcal{T}_i} \frac{\tilde{x}_i(t) + \delta(t)}{p_i(t)}$$

$$= \sum_{t \in \mathcal{T}_i} \frac{\tilde{x}_i(t)}{p_i(t)} + \sum_{n \in [N_i]} \frac{\gamma}{q_i(t_i(n))} \left( 1 - \frac{q_i(t_i(n))}{q_i(t_i(n-1))} \right) + \sum_{n \in [N_i]} \left( \frac{1}{p_i(t_i(n))} - \frac{1}{q_i(t_i(n))} \right)$$

$$= \sum_{t \in \mathcal{T}_i} \frac{\tilde{x}_i(t)}{p_i(t)} + \sum_{n \in [N_i]} \gamma \left( \frac{1}{q_i(t_i(n))} - \frac{1}{q_i(t_i(n-1))} \right) + \sum_{n \in [N_i]} \left( \frac{1}{p_i(t_i(n))} - \frac{1}{q_i(t_i(n))} \right)$$

$$= \sum_{t \in \mathcal{T}_i} \frac{\tilde{x}_i(t)}{p_i(t)} + \frac{\gamma}{q_i(t_i(N_i))} + \sum_{n \in [N_i]} \left( \frac{1}{p_i(t_i(n))} - \frac{1}{q_i(t_i(n))} \right). \quad (13)$$

Now, assuming $\gamma \geq \Phi$, we have

$$\sum_{t \in [T]} \hat{x}_i(t) = \sum_{t \in \mathcal{T}_i} \frac{\tilde{x}_i(t)}{p_i(t)} + \frac{\gamma}{q_i(t_i(N_i))} + \sum_{n \in [N_i]} \left( \frac{1}{p_i(t_i(n))} - \frac{1}{q_i(t_i(n))} \right)$$

$$= \sum_{t \in \mathcal{T}_i} \frac{\tilde{x}_i(t)}{p_i(t)} + \frac{\gamma - \Phi}{q_i(t_i(N_i))} + \frac{\Phi}{q_i(t_i(N_i))} + \sum_{n \in [N_i]} \left( \frac{1}{p_i(t_i(n))} - \frac{1}{q_i(t_i(n))} \right)$$

$$\geq \sum_{t \in \mathcal{T}_i} \frac{\tilde{x}_i(t)}{p_i(t)} + \frac{\gamma - \Phi}{q_i(t_i(N_i))} + \frac{1}{q_i(t_i(N_i))} \sum_{t \in \mathcal{T}_i} |a(t)| + \sum_{n \in [N_i]} \left( \frac{1}{p_i(t_i(n))} - \frac{1}{q_i(t_i(n))} \right).$$

Using $q_i(t_i(N_i)) \leq q_i(t)$ for any $t$, and rewriting some terms in the above equation, we have

$$\sum_{t \in [T]} \hat{x}_i(t) \geq \sum_{t \in \mathcal{T}_i} \frac{\tilde{x}_i(t)}{p_i(t)} + \frac{\gamma - \Phi}{q_i(t_i(N_i))} + \sum_{t \in \mathcal{T}_i} \frac{|a(t)|}{q_i(t)} + \sum_{t \in \mathcal{T}_i} \left( \frac{1}{p_i(t)} - \frac{1}{q_i(t)} \right). \qquad (14)$$

In view of $0 \leq |a(t)| \leq 1$, the last two terms in the right-hand side satisfy

$$\sum_{t \in \mathcal{T}_i} \frac{|a(t)|}{q_i(t)} + \sum_{t \in \mathcal{T}_i} \left( \frac{1}{p_i(t)} - \frac{1}{q_i(t)} \right) \geq \sum_{t \in \mathcal{T}_i} \frac{|a(t)|}{q_i(t)} + \sum_{t \in \mathcal{T}_i} |a(t)| \left( \frac{1}{p_i(t)} - \frac{1}{q_i(t)} \right)$$

$$= \sum_{t \in \mathcal{T}_i} \frac{|a(t)|}{p_i(t)} \geq \sum_{t \in \mathcal{T}_i} \frac{a(t)}{p_i(t)}$$

Putting this together with the fact that $q_i(t_i(N_i)) \leq 1/K$, we thus the desired result:

$$\sum_{t \in [T]} \hat{x}_i(t) \geq \sum_{t \in \mathcal{T}_i} \frac{\tilde{x}_i(t)}{p_i(t)} + \frac{\gamma - \Phi}{q_i(t_i(N_i))} + \sum_{t \in \mathcal{T}_i} \frac{a(t)}{p_i(t)} \geq \sum_{t \in \mathcal{T}_i} \frac{x_i(t)}{p_i(t)} + (\gamma - \Phi)K \,.$$

$\square$

## D   Numerical Results

(a) Empirical Regret of `ExpRb` and `Exp3` Under Attacks

(b) Performance Comparison Under Different Corruption Levels

Figure 1: Experimental results

In the simulation, we evaluate the performance of `ExpRb` and compare it with the `Exp3` algorithm in different scenarios. In order to evaluate our algorithm under adversarial corruptions, we assume the attacker follows the so-called attack-optimal-arms policy introduced in Section 2. The attack-optimal-arms policy can efficiently attack the empirical reward estimation of the optimal arm and trick the learning algorithm to select a suboptimal arm.

The experimental scenario is as follows. Consider an environment with one high-reward arm and $K - 1$ low-reward arms. The attacker aims to decrease the observed reward when the online learner chooses the optimal arm. When the budget is available, the attacker will always set the reward on the optimal arm to be zero when it was chosen. We report the average regret obtained by collecting the actual regret of 100 execution of this scenario. The first scenario involves an attacker with attack budget of $O(\sqrt{T})$. The simulation results shown in Figure 1(a) imply that the performance of the `Exp3` algorithm is largely degraded with the attack. The `ExpRb` algorithm, however, achieves a sublinear regret. In the second scenario, we compare the performance of two algorithms under different amount of budget of attacker. Toward this, we vary the available budget of the attacker in 10 levels. The corresponding budget for the $l$-th level is $T^{0.2+l/20}$. Figure 1(b) shows the performance comparison between the `Exp3` algorithm and the `ExpRb` algorithm. One can find that the performance of `Exp3` is largely degraded when the attacker budget reaches $T^{1/2}$, while the `ExpRb` algorithm can tolerate heavier attacks.