[Reviews · NeurIPS 2020]

Review 1

Summary and Contributions: This paper studies adversarial bandits with corruptions, where at each round, after the adversary commits to a reward vector and the learner selects an action, an attacker can corrupt the reward of the action taken by the learner, and use this to give learner misleading information. Before this work, bandits with corruption is only studied in the stochastic setting. The paper shows a set of positive and negative results: 1. If the learner does not have access to an upper bound on the adversary's corruption budget, then the attacker can force the learner to have Omega(T) regret with a sublinear corruption budget. 2. If the learner uses EXP3, the adversary can force the learner to have Omega(T) regret with only O(\sqrt{T}) corruption budget. 3. In the setting where the learner knows the attacker's corruption budget \Phi: - On the negative side, any learner need to suffer at least Omega(\sqrt{T} + \Phi) under some environment - On the positive side, an algorithm ExpRb is proposed that achieves an matching regret upper bound of O(\sqrt{T} + \Phi) up to log factors.

Strengths: - The results are novel: the first paper that studies adversarial bandits with corruption - The knowledge of attacker's budget \Phi is well-motivated, in view of a fundamental lower bound (Theorem 1) - In the attack-aware setting, the paper establishes matching regret upper and lower bounds.

Weaknesses: Perhaps the only weakness I can see about this paper is some presentation issues. Specifically, although I can follow most of the proofs step-by step, it is a bit hard for me (as an non-expert) to appreciate why this paper uses this particular technique. For example, has similar biased estimator (Equation 5) been proposed before in the literature? Also, why does \delta(t) has the specific functional form in (7)?

Correctness: I have checked most of the proofs, but there are a few points that I am still confused and would appreciate the author's clarification: 1. In line 541, case (i), why can we directly use \tilde{p}_i(t_i(n)) = p_i(t_i(n))? Wouldn't it be possible that we fall into case (1) in line 522? 2. In line 551, do we need p_i(t) <= \tilde{p}_i(t)?

Clarity: It is well written. I appreciate that the authors gave a proof sketch of Theorem 4.

Relation to Prior Work: Yes. See "strengths" above.

Reproducibility: Yes

Additional Feedback: After reading the rebuttal: I thank the authors for clarifying the definition of T_i - the proof now goes through as far as I have double checked. It would be useful to mention \hat{x}_i(t) = I(I_t = i) x_i(t) / p_i(t) when (1) I_t != i or (2) I_t = i and delta(t) = 0 so that the connection between Lemma 8 and 6 becomes transparent. Additional minor comments: - in Lemma 8 the right hand side should have an extra c(\gamma - \phi) K as in Lemma 6. - I found the max and min notation in Algorithm 1 confusing. I personally would prefer listing the three cases depending on the comparison results between p_{I_t}(t) / \tilde{p}_{I_t} and 1-1/gamma, and 1. - Perhaps the proof of Theorem 1 is incomplete (although I think is fixable): What happens if the algorithm selects arm l for Theta(T/ln T) times? Using the existing proof, we can show that either it has Omega(T/ln T) regret in environment 1 (with no corruption) or it has Omega(T) regret in environment 2 (with Omega(T/ln T) corruption).


Review 2

Summary and Contributions: This paper studies an adversarial bandit setting where the attacker can attack the returned reward after observing the action selected by the agent in each round. The paper first proves that all bandit algorithms will fail when the budget of attacker is unknown to the agent, then proposes a novel bandit method introducing biased estimator to mitigate the risks of underestimation and overestimation of the actual reward, finally, a near optimal regret bound is proved.

Strengths: 1. the theoretical analysis is solid. The lower bound and upper bound are both proved, and the proposed algorithm is near optimal. 2. the problem is general in practice, and this paper is the first to solve the problem.

Weaknesses: 1. there is no empirical experiments.

Correctness: correct

Clarity: well written

Relation to Prior Work: Yes

Reproducibility: Yes

Additional Feedback:


Review 3

Summary and Contributions: The paper studies adversarial or non-stochastic bandits with corruptions by an attacker who is able to corrupt the reward after the action of the learner is decided. The authors first prove an impossibility result that no algorithm that is unaware of the existence of the attacker is able to guarantee sub-linear regret even if the budget of the attacker is sub-linear. Second, the authors show a lower bound for any algorithm that is aware of the existence and budget of the attacker, and gives a robust algorithm that achieves the lower bound up to logarithmic factors.

Strengths: The paper is easy to follow. I found the related work section in the supplementary material well-written and helpful for understanding contributions of this paper and other related papers. The theoretical claims look correct even though I didn’t check all the proofs in the appendix. The paper is clearly very relevant for NeurIPS community. With respect to significance and novelty, my understanding is that the main novelty lies in the study of corruption under the adversarial bandit instead of stochastic bandit setting.

Weaknesses: When I first read the definition of the targeted attack, I find it hard to think of an example in reality. I think a few more sentences on the motivation behind this definition and its tie to reality would help the readers understand the potential significance of this problem. The main novelty lies in the study of adversarial instead of stochastic bandits, which is limited in my opinion. But I am no expert in this field so my judgement could be wrong.

Correctness: The methods and claims look correct, but I didn’t check all the proofs in the appendix.

Clarity: The paper is well-written.

Relation to Prior Work: The supplementary material clearly discussed the difference between this work and previous contributions.

Reproducibility: Yes

Additional Feedback: I have two questions for the authors: 1. Theorem 1 states that for any attack-agnostic algorithm, there is any attacker with sub-linear budget that can force linear regret, is it possible to give a more refined upper bound on the necessary budget for any algorithm? For example, Corollary 2 says O(\sqrt{T}) budget is necessary to attack Exp3, is O(\sqrt{T}) enough for other algorithms? 2. Is it possible to give an adaptive algorithm that can nearly achieve the optimal regret O(\sqrt{T} + \Phi) without knowing the exact budget (but knowing the existence of a targeted attacker)? Or alternatively, give a lower bound showing this is impossible. -------------------- The author's response sufficiently addressed my questions and concerns, hence I decided to raise my score.


Review 4

Summary and Contributions: Considers the adversarial bandit setting where an attacker can corrupt rewards after observing an action of the learner. The paper shows that algorithms that are agnostic to the attackers corruption budget cannot obtain sub-linear regret, and provide an algorithm, ExpRb, that uses adjusted reward estimates to lead to logarithmic regret if the budget is known.

Strengths: - Studies a novel setting on corruption in MABs where the attacker manipulates rewards after observing actions - The work addresses this setting holistically by providing an impossibility result for agnostic algorithms, and providing both lower bounds and an attacker aware algorithm with regret guarantee - The paper does a great job giving intuition around the issues that arise in the particular setting and how to get around them.

Weaknesses: - Lacking a bit of motivation why non-stochastic bandits with corruptions (jointly) are interesting to study. It argues corruptions are interesting, it argues adversarial setting is interesting, but both in isolation and does not explain why the combination is relevant (individual settings have already been studied). - The impossibility theorem (1) is rather weak: it only claims that there exists a budget (it may need to be very large) that can force linear regret. A stronger bound could show how regret deteriorates as a function of the budget, or show that even with a small budget, the attacker can force linear regret (the corollary is a great example of this) - The algorithm needs to know the attackers budget (or at least an upper bound); this makes it less relevant from a practical point of view.

Correctness: I have found no issues.

Clarity: The paper does a great job explaining the reasoning and is easy to follow for the most part. However, sometimes it's more challenging to follow the reasoning. E.g. in lines 177-180 I find it difficult to parse the argument.

Relation to Prior Work: The paper discusses prior work in detail, both in the main text and appendix.

Reproducibility: Yes

Additional Feedback: - The paper focuses on the case where the actual reward is corrupted, rather than the observation. While the authors note that results can easily be translated between the two, I find the latter more natural and wonder why the work doesn't focus on that instead Overall, the paper introduces a new setting, but the work doesn't provide a strong argument for its relevance. Both the impossibility result and guarantees of the robust algorithm are a positive contribution to the field, but it's unclear that this is a significant step for the wider Neurips community; the impossibility result in its general form is quite weak and the ExpRb algorithm is very much an adaptation of existing techniques. In particular, it's unclear how future work would see this as a fundamental contribution on MABs with corruption. Minor remarks: - Typo on line 60: attack-agnostic - Typo on line 72: this has not conflict to - Line 90: this claim is a bit confusing here without reading the paper further: gamma needs to be Phi so one does need to know Phi? POST REBUTTAL Thanks for the detailed clarifications to comments raised by the reviewers. Based on the response and our discussion, I've increased my score to a 7.

[Author Response · NeurIPS 2020]

We thank the reviewers for their insightful comments. The numbered citations refer to references in the submitted paper. The remaining additional citations are listed at the end of the page.

**Reviewers 3 and 4: motivation.** We admit the comment of **Reviewer 4** about the lack of motivation on *the combined scenario of nonstochastic bandits with corruption*. The isolated motivation about bandits with corruption is mentioned in 2nd paragraph of the introduction (L25-35) with extensive recent results [8,9,11,14,15,17,18,25]. The motivation about nonstochastic bandits is given in the 3rd paragraph (L36-43). There exist application scenarios where either 'nonstochastic bandits' or 'bandits with corruption' in insolation fail to fully characterize the underlying model. To response *the combined motivation* and strengthen the motiviation, we will add one such example, described below, to the paper. This will also address **Reviewer 3**'s comments on *the real example of targeted attacks* and **Reviewer 4**'s comments on attacks to the *actual vs. observation of the reward.*

A concrete example of nonstochastic bandits with corruptions is the Online Shortest Path Routing (OSPR) problem under the denial of service (DoS) attacks. OSPR is a classic example of MAB problems [20]. And there is also extensive research on routing under DoS attacks, including the recent work [Zhou et al., 2019] focusing on bandit modeling of this scenario. OSPR could be reasonably modeled as nonstochastic bandits when the delays on the links change dynamically [György et al. 2007], or once is it difficult to characterize the combined distribution of a path including multiple links [20]. In this nonstochastic scenario, the DoS attack could be modeled by our bandit with targeted corruptions. Specifically, the DoS attacker can be aware of the selected paths by detecting the transmitted packets over the path and manipulate the latency of the selected path by flooding the path with dummy packets. Also, the budget of the attacker is simply the available resources for the DoS attacker to keep her undetectable. Arguably, none of 'nonstochastic bandits' and 'bandits with corruption' alone would suffice to fully characterize the underlying model here. We believe presenting this example in the introduction would prove useful to connect the dots between *nonstochastic bandits* and *bandits with corruption*.

**Reviewer 2: experiments.** We will add the following experiment to the supplementary material. The goal here is to compare ExpRb with Exp3. We constructed a simple scenario where the attacker follows a simple policy that attacks the optimal arms (see L172-181 of the paper) with 1 high-reward and $K-1$ low-reward arms. In Fig. 1, we report the average regret of 100 independent runs, with $\Phi = O(\sqrt{T})$. The results show that the regret of Exp3 is largely degraded with the attack, while ExpRb achieves a sublinear regret. This experiment is not meant to be exhaustive, rather it is intended to validate the theoretical results and illustrate the potential of our approach.

Figure 1: ExpRb vs. Exp3

**Reviewers 3 and 4: lower bound. R3/Q1:** Different bandit algorithms tolerate different levels of corruption. Hence, finding a more refined bound for the attacker's budget is highly algorithm-specific and we were not able to generalize our result in Corollary 2 for algorithms beyond Exp3. **R3/Q2:** Without substantial change in the proofs, the lower bound in Theorem 1 can be applied to the algorithms that are aware of the existence of corruptions, but do not know the budget. We will add a remark after Theorem 1 to involve this case. **R4 (the impossibility Theorem 1):** The impossibility result is a generic result for any attack-agnostic algorithms; one can have more refined versions for specific algorithms such as our result for Exp3 (see **R1/Q1**). **R4: the algorithm needs to know the attacker's budget**): Yes, it is a negative result on the attack-agnostic model. ExpRb algorithm is parameterized by a robustness parameter $\gamma$. Our further results which will appear in the future version can characterize the regret as a function demonstrating the regret reduction with improper $\gamma$. As future work, it is promising to dig out more interesting results on the attack-agnostic case.

**Additional comments: Reviewer 1:** Both questions are valid and due to a mistake when defining $\mathcal{T}_i$, $t_i(n)$ and $N_i$ in the proof of Lemma 8. Thanks for your careful reading. $\mathcal{T}_i$ should be referred to (Do you simply mean: "$\mathcal{T}_i$ denotes") the set of time slots that the $i$-th arm is selected and the selection probability for arm $i$ is lower than the previous one (only in this case, $\delta(t)$ will be larger than 0). And accordingly, $t_i(n)$, $n = 1, 2, \ldots, N_i$ are the indices for those time slots. By redefining those variables, we hope we can clarify the reviewer's concern as follows. **R1/Q1:** We only consider the time slots that the selection probability for the $i$-th arm is smaller than the previous, so the algorithm falls into the case in Line (5) of algorithm 1, but not case (1) in line 522. By checking the conditions for Equation (8), we have $\tilde{p}_i(t_i(n)) = p_i(t_i(n))$. **R1/Q2:** Yes, we do require that condition. This inequality holds, since we only consider the time slots that the selection probability for the $i$-th arm is smaller than the previous one.

**Additional References**

[Zhou et al., 2019] Zhou, P., Xu, J., Wang, W., et al. (2019). Toward Optimal Adaptive Online Shortest Path Routing With Acceleration Under Jamming Attack. *IEEE/ACM Transactions on Networking*, 27(5), 1815-1829.

[György et al. 2007] György, A., Linder, T., Lugosi, G., and Ottucsák, G. (2007). The on-line shortest path problem under partial monitoring. *Journal of Machine Learning Research*, 8(Oct), 2369-2403.


[Meta-Review · NeurIPS 2020]

Reviewers are satisfied with the rebuttal, and two of them increased their score. Several promises were made in the rebuttal, so please do make sure that these are incorporated in the final version.